# Maxillary incisor enamel defects in individuals born with cleft lip/palate

**Juliane R. Lavôr[1], Rosa Helena W. Lacerda[1], Adriana Modesto[2], Alexandre R. Vieira[1,2]***

**1** Graduate Program in Dentistry, Universidade Federal da Paraíba, João Pessoa, Brazil, **2** Departments of Pediatric Dentistry and Oral Biology, School of Dental Medicine, University of Pittsburgh, Pittsburgh, Pennsylvania, United States of America

* arv11@pitt.edu

**Data Availability Statement:** All relevant data are within the manuscript and its Supporting information files.

**Funding:** The author(s) received no specific funding for this work.

## Abstract

Cleft lip with or without cleft palate (CLP) is considered the most frequent congenital malformations of the head and neck, with cleft individuals exhibiting more chances of presenting abnormalities such as developmental defects of enamel (DDE). Matrix metallopeptidase 2 (*MMP2*) is a membrane-bound protein with collagen-degrading ability and has important roles in tooth formation and mineralization. The aim of this study was to evaluate the frequency, location, severity and extent of DDE found in the maxillary incisors for groups of individuals born with CLP, as well as understanding their relationship with the cleft side. Besides, this study addresses the hypothesis that DDE can be influenced by variation in the *MMP2* genes (rs9923304). Individual samples, clinical history, intraoral photographs and panoramic radiographs were obtained from 233 patients under treatment at the Cleft Lip and Palate Service of the University Hospital Lauro Wanderley at the Federal University of Paraíba. Digital images were examined by the same evaluator using the Classification of Defects According to the Modified DDE Index, and then loaded into the Image Tool software, where two measurements were made: total area of the buccal surface (SA) and the area of the DDE (DA), obtaining the percentage of the surface area affected (%SAD) (ICC = 0.99). Genomic DNA was extracted from saliva samples from 124 participants. Genotyping was carried out using TaqMan chemistry for one marker in *MMP2* (rs9923304). Statistical analyses were performed by The Jamovi Project software. The Shapiro-Wilk test was applied, followed by the Student's t-test and the Mann-Whitney test. Chi-square and Fisher's exact tests, and odds ratio (OR) with 95% confidence interval (CI) calculations were used to determine Hardy-Weinberg equilibrium and statistically significant differences with an alpha of 0.05. No significant differences in the prevalence and extent of enamel defects were found between male and female individuals born with CLP (p = 0.058256). The frequency of individuals presenting teeth with DDE, in relation to the cleft and non-cleft side, was statistically different (p <0.001; OR = 7.15, CI: 4.674> 7.151> 10.942). However, the averages of %SAD were similar (p = 0.18). The highest means of the %SAD were found in individuals with bilateral cleft lip with or without cleft palate (BCLP) when compared to individuals with unilateral cleft lip with or without cleft palate (UCLP), for the teeth inside (IA) and outside the cleft area (OA) (p <0.001). Regardless of the cleft side, individuals with BCLP were 7.85 times more likely to have more than one third of the tooth surface affected, showing more

**Competing interests:** The authors have declared that no competing interests exist.

frequently defects in the three thirds (OA: p <0.001) (IA: p = 0.03), as well as a higher frequency of more than one type of defect (OA: p = 0.000358) (IA: p = 0.008016), whereas in UCLP, defects were isolated and restricted to only one third, more frequently, the incisal third (OA: p = 0.009) (IA: p = 0.001), with greater frequency of milder defects, such as demarcated (p = 0.02) and diffuse (p = 0.008) opacities. A higher frequency of the T allele, less common, was observed in the group of CLP individuals who had all the affected teeth or at least two teeth with %SAD greater than 20% (p = 0.019843). Our results suggest that *MMP2* may have a role in the cases that presented DDE and genotyping rs9923304 could serve as the basis for a genomic approach to define risks for individuals born with CLP. Frequency and severity of DDE is strongly related to the CLP phenotype, since the highest values were found for BCLP. However, the extent of the DDE is independent of its relationship with the side of the cleft.

## Introduction

Cleft lip and palate are a group of structural malformations around the oral cavity and can extend on to the face resulting in oral and facial deformities, being considered the most frequent congenital malformations of the head and neck [1]. It has a complex etiology, with genetic and environmental factors and their interactions playing an important role [1, 2], with an average birth prevalence of 1/700 [1, 3, 4]. By tradition, oral clefts have been referred to as cleft lip with or without cleft palate (CLP) and cleft palate only (CPO), due to differences in embryology [1].

Overall, 15% of all oral clefts are syndromic, and they are part of more than 300 recognized syndromes. Of the remaining 85% of individuals with non-syndromic cleft, 50% have other less well-defined anomalies [4, 5], with individuals born with clefts exhibiting at least four times more chances of presenting abnormalities such as agenesis, supernumerary teeth [6, 7], enamel defects and/or microdontia [8, 9].

The severity of the dental anomalies seems to be directly related to the severity of the cleft [6, 10, 11], suggesting that the embryological development of the lip, palate and tooth is controlled by common genetic factors [3]. There is, however, a gap in the knowledge regarding the reason why there is a correlation between the type of cleft and the severity of the dental abnormalities found. Many studies attribute the phenotype of oral clefts simply to 'affected' or 'unaffected' status, while evidence increasingly indicates that other clinical markers, such as the presence of dental anomalies, should be considered, defining broader phenotypes that help to unravel the genetic basis of the condition [6, 12].

Individuals born with oral clefts have a higher frequency of developmental defects of enamel (DDE) [7, 8, 10], faulty or deficient formations of enamel on primary and permanent teeth during tooth development, resulting in hypoplasia, a quantitative defect, and/or hypomineralisation, namely opacity, a qualitative defect characterized by abnormal enamel translucency [13–15].

The hypomineralized enamel is softer and more porous, facilitating the accumulation of plaque and the development of dental caries [16]. Therefore, DDE represent a risk indicator for dental caries and erosive tooth wear in children [14, 15].

In both dentitions, enamel defects are found more frequently in the maxillary incisors [7, 8, 10]. It is believed that the teeth adjacent to the cleft show more pronounced changes, in

comparison with the nonclefted side [8, 17]. There are few studies available in the literature that compared the prevalence of enamel defects between the affected and unaffected sides of individuals born with CLP [8, 17–19]. However, none of them were able to measure the extent and, therefore, the severity of these defects.

The matrix metalloproteinases (MMPs) constitute a family of secreted and membrane-associated zinc-dependent endopeptidases that are the major regulators of extracellular matrix turnover and are capable of selectively degrading a wide spectrum of both extracellular matrix and non-matrix proteins [20, 21]. Matrix metalloproteinase 2, also known as gelatinase A, is a membrane-bound protein with collagen-degrading ability [20, 22] and has important roles in tooth formation and mineralization [23–27]. Genetic polymorphisms in enamel formation genes could contribute to structural alterations that could lead to enamel porosity, presence of enamel crystal inhibitory proteins and decreased mineral content [21, 28, 29]. Some studies have demonstrated the expression of MMP2 during enamel formation [23–25]. However, no study has investigated polymorphisms in *MMP2* in enamel defects.

Thus, the aim of this study was to investigate the characteristics of enamel defects in individuals born with oral clefts. We evaluated the frequency, location, character of structural changes in dental morphology, and severity and extent of enamel defects found in the maxillary incisors for groups of individuals born with CLP, as well as attempted to understand their relationship with the cleft side. Besides, this study addressed the hypothesis that DDE can be influenced by variation in the *MMP2*.

## Materials and methods

This is a cohort cross-sectional study approved by the Research Ethics Committee of the University Hospital Lauro Wanderley of the Federal University of Paraíba [CAAE 13450819.6.0000.5183]. All 233 individuals born with CLP without any other structural abnormalities and with suitable materials were evaluated. The sample had a mean age of 13.13 years (ranging from 6 to 35 years-old). Written informed consent documents were obtained from all subjects. We followed the STREGA guidelines for this study. Photographic material was obtained prior to orthodontic treatment and several years after surgical lip repair.

Individual samples, clinical history, intraoral photographs and panoramic radiographs were obtained from patients under treatment at the Cleft Lip and Palate Service of the University Hospital Lauro Wanderley at the Federal University of Paraíba. The exclusion criteria included labial surfaces of permanent central and lateral incisors not accessible for proper examination (presence of restorations, orthodontic appliances or crowns) or individuals with low quality intraoral photographs.

The methodology of the study was divided into five stages:

### Stage 1—Intraoral photographs

Standardized photographs were taken by an experienced specialist (R.H.W.L), with the patient sitting in a dental chair and leaning back to avoid movement during focus and photography. After an initial examination, the surfaces of the teeth were cleaned and dried and the appearance of the enamel was recorded using a digital camera (Canon EOS Rebel T5i, Ohta-ku, Tokyo, Japan), with standard lens (Canon EF 100 mm macro lens) and settings (ISO 6400, speed 1/125 and aperture F/25), always under the same flash (Macro Ring Flash Sigma EM-140 DG) and natural lighting conditions. Cheeks and lips were retracted using T-Shape intraoral cheek lip retractor, which was sterilized prior to each use. The photographs were taken focusing on the center of the four permanent incisors [30]. Two photographs with side

views were also taken, each showing the lateral incisors and canines on each side of the arch [31], which were used when doubts arose when analyzing the frontal photograph. Each photo was evaluated for acceptability and quality by the photographer (R.H.W.L.). When a photograph was not acceptable because of being out of focus, it was repeated.

## Stage 2—Determination of cleft and DDE phenotypes

The determination of the cleft phenotype was obtained through information from the medical record. It was based on the extent and laterality of the cleft through a single examiner with more than 10 years of experience in examining tooth surfaces of children born with oral clefts (R.H.W.L).

To eliminate inter-examiner differences, existing intraoral photographs of all participants were examined by one single evaluator (J.R.L), after being calibrated by an experienced specialist (R.H.W.L), using the Classification of Defects According to the Modified DDE Index based on the Fédération Dentaire Internationale recommendation [32]. It allowed classification of participants to any of the following diagnostic subgroups: (0) normal; (1) demarcated opacity; (2) diffuse opacity; (3) hypoplasia; (4) other defects; and its combinations: (5) demarcated and diffuse; (6) demarcated and hypoplasia; (7) diffuse and hypoplasia; (8) all three defects. A new code (9) was added due to the inability of observing defects in some teeth, especially those adjacent to the cleft, where the tooth is often distalized, mesialized, ectopic, not erupted or absent (Table 1). Corresponding panoramic radiographs were also analyzed.

## Stage 3—Measuring the extent of enamel defects: Proposal for a new method

Digital images were loaded into the Image Tool software (v. 3.0, San Antonio Dental School, University of Texas Health Science, TX, USA). Two measurements were made: (1) the proportion of the total area of the buccal surface in the photograph (SA); (2) and the area of each enamel defect found in the respective tooth (DA), thus obtaining a ratio that indicated the percentage of the surface area affected by the DDE ($\%SAD = (DA)/(SA) \times 100$).

A pilot study was carried out with 20 patients to identify possible errors, test, and validate the proposed method. Three weeks later, the first assessment was carried out, with all the necessary adjustments raised by the pilot made. The intra-examiner agreement was assessed by a

**Table 1. Classification of defects that includes a modification of the modified DDE index, which includes the addition of code 9.**

| Defect | Code |
|---|---|
| Normal | 0 |
| Demarcated opacity | 1 |
| Diffuse opacity | 2 |
| Hypoplasia | 3 |
| Other defects | 4 |
| *Combinations* | |
| Demarcated and diffuse | 5 |
| Demarcated and hypoplasia | 6 |
| Diffuse and hypoplasia | 7 |
| All three defects | 8 |
| Unobservable or absent | 9 |

**Table 2. Details of the gene and SNP investigated in this study.**

| Chromosome | Gene | SNP marker | Base Position | SNP Function | Base change [a] |
|---|---|---|---|---|---|
| 16 | matrix metallopeptidase 2 (MMP2) | rs9923304 | 55496389 (GRCh38.p12) | Intron | CT |

[a]Ancestral allele listed first.

second evaluation of the images after three weeks, with an Intraclass Correlation Coefficient (ICC) of 0.999.

## Stage 4—DNA samples and genotyping

Genomic DNA was extracted from saliva samples from 124 participants, using established protocols [33]. A single nucleotide polymorphism (SNP) in the *MMP2* gene (rs9923304) was selected, considering disequilibrium linkage and gene structure. This SNP was selected based on published reports and its location within the gene. We used information from the NCBI dbSNP (https://www.ncbi.nlm.nih.gov/snp/) and HUGO Gene Nomenclature Committee (https://www.genenames.org/tools/multi-symbol-checker/) databases. Details of the selected gene and polymorphism are presented in Table 2.

Genotyping was performed by polymerase chain reactions using the Taqman method [34] with an ABI PRISM QuantStudio 6 Flex automatic instrument and pre-designed probes (Applied Biosystems, Foster City, CA, USA).

## Stage 5—Statistical analysis

Data processing and statistical analysis were performed by The Jamovi Project software (Version 1.1). The sample size included the spontaneous demand of patients at the Cleft Lip and Palate Service of the University Hospital Lauro Wanderley, respecting the inclusion criteria. In order to verify the normal distribution of the numerical variables, the Shapiro-Wilk test was applied, followed by an analysis of variance with the Student's t-test and the Mann-Whitney test, in the cases of normal and non-normal distribution, respectively.

Chi-square and Fisher's exact tests, and odds ratio (OR) with 95% confidence interval (CI) calculations were used to determine Hardy-Weinberg equilibrium and statistically significant differences with an alpha of 0.05.

## Results

A total of 233 individuals born with cleft lip with or without cleft palate (CLP) were evaluated. Of this total, 105 were female and 128 were male, with an average age of 13.13 (ranging from 6 to 35 years).

Through the proposed method (Figs 1–3) it was possible to obtain the percentage of the buccal tooth surface area affected by the defect of each unit, with an average Intraclass Correlation Coefficient (ICC) of 0.999, resulting from the reassessment of 20 patients after six weeks of the pilot study. ICC $\geq$ 0.75 confirms the excellent reproducibility of a study.

Taking into account the maxillary permanent central and lateral incisors of each patient, 681 teeth were evaluated, of which 469 exhibited developmental defects of enamel (DDE), while only 195 were unaffected. Of the affected teeth, 53% were from male individuals, with an average of the surface area affected by the defect (%SAD) of 44%, while 46% were from female individuals, with an average %SAD of 39.75% (p = 0.058256). Of these, 270 were inside the cleft area, while 199 were outside the cleft area.

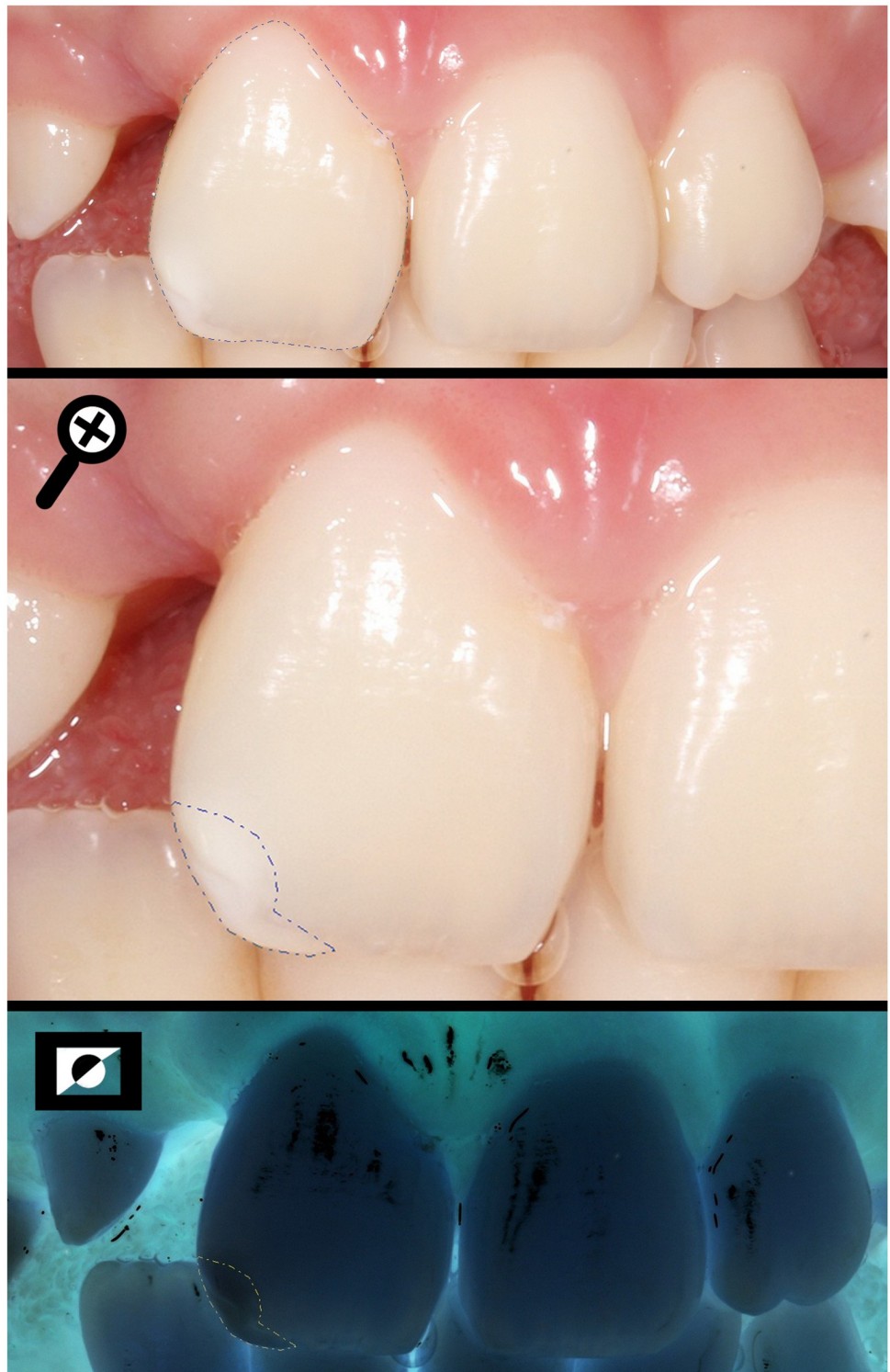

**Fig 1. Measuring the total area of the buccal surface (SA) and the area of the enamel defect found in the incisal third (DA) of a UCLP individual, using the negative photo as a tool to help to limit the extension.**

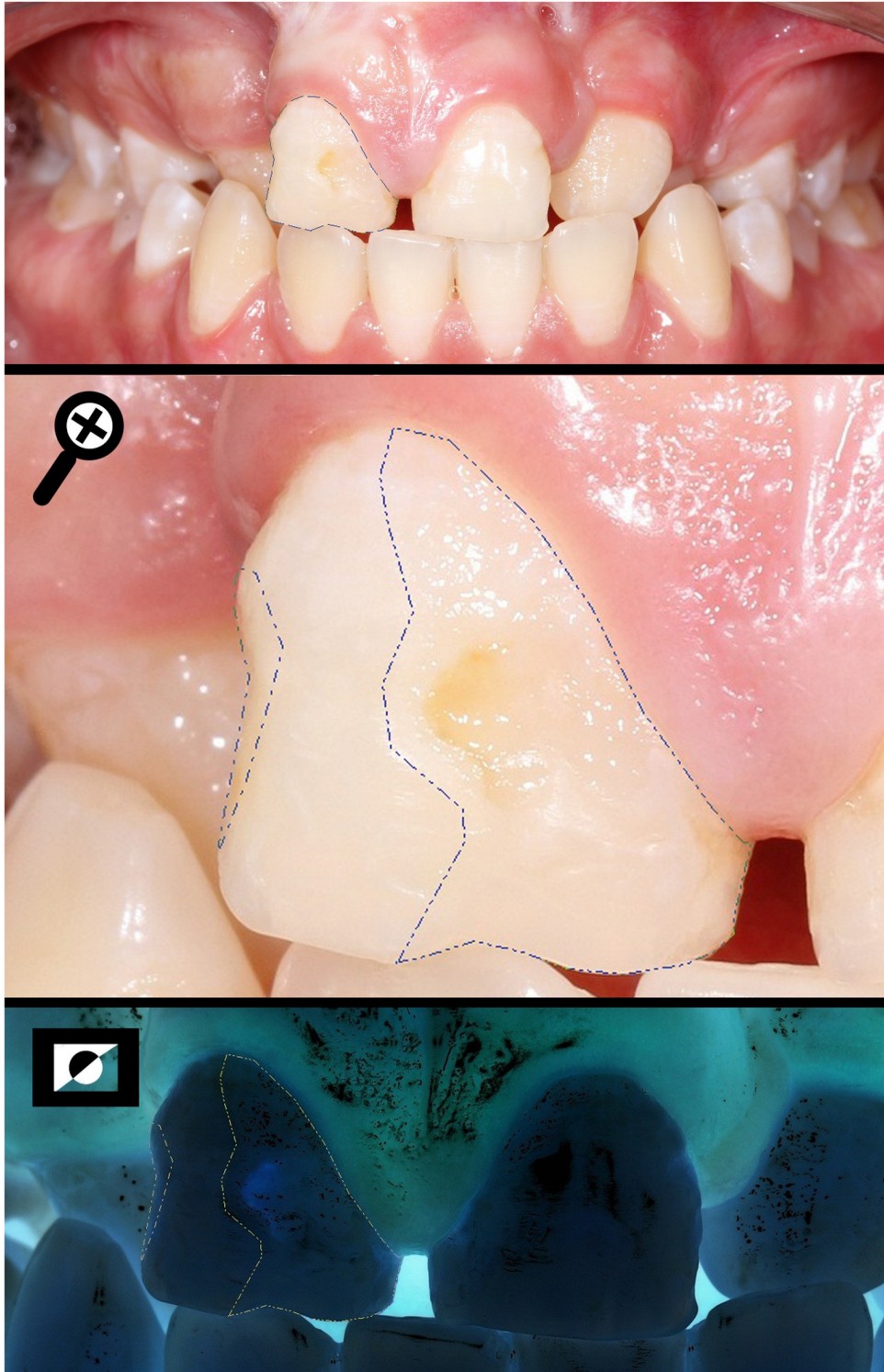

**Fig 2. Measuring the total area of the buccal surface (SA) and the area of the enamel defect found in all three thirds (DA) of a BCLP individual.**

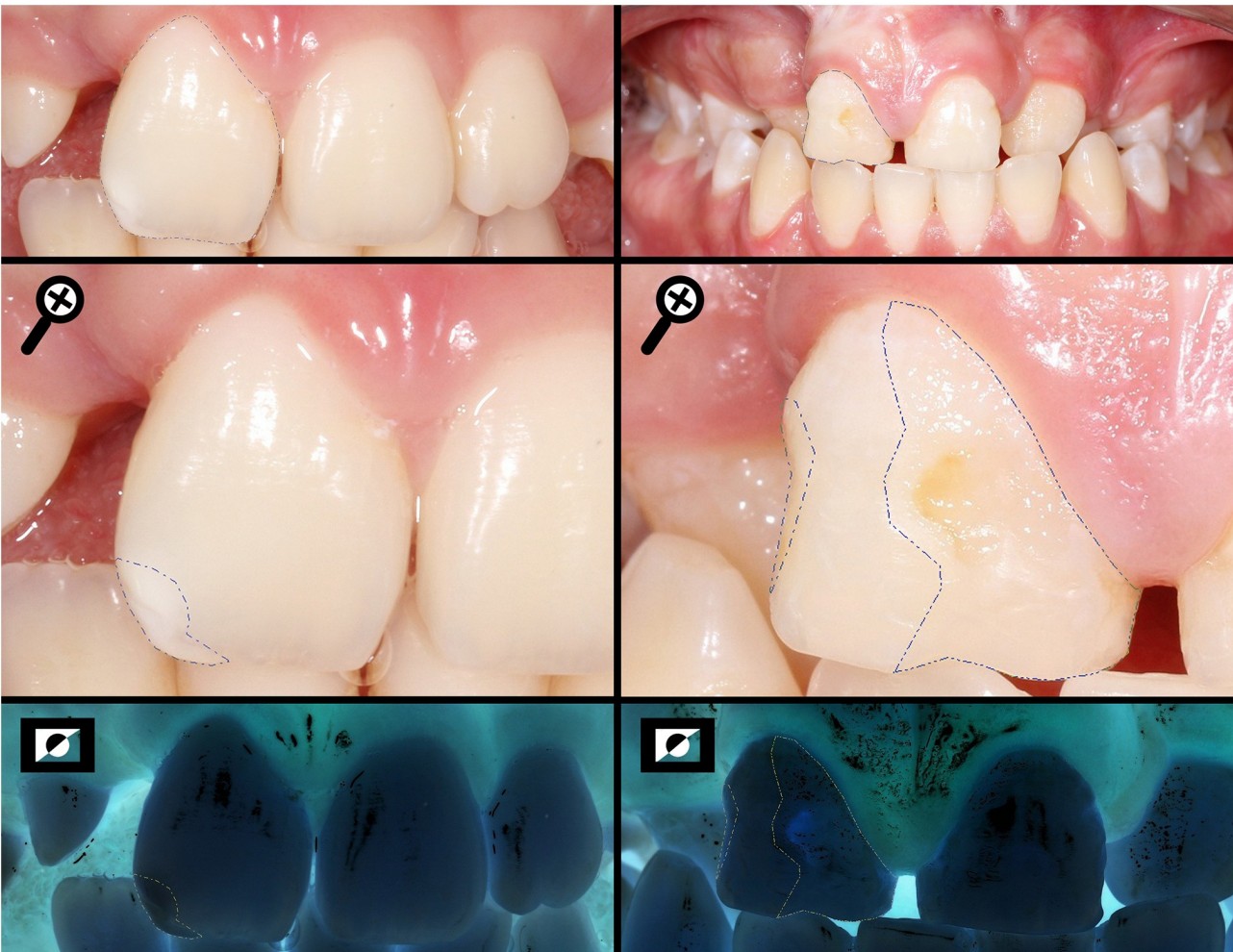

**Fig 3. Measuring the total area of the buccal surface (SA) and the areas of the enamel defects found (DA) of a UCLP and BCLP individuals, respectively, using the negative photo as a tool to help to limit the extension.**

Statistically significant differences (p<0.001) were found when comparing bilateral and unilateral clefts in relation to the percentage of the surface area affected by the defect of the teeth outside the cleft area, with the highest mean and median values observed in the cases born with bilateral clefts, as shown in Table 3.

Regarding the percentage of the surface area affected by the defect (proposed method) and the modified DDE index [32], statistically significant differences (p <0.001) were observed for the teeth outside the cleft area, showing a low Spearman's correlation coefficient (r = 0.2748), resulting from the difference between the variables: while the first is continuous, the second is ordinal.

The frequency of the types of enamel defects concerning their location on the buccal surface showed a significant difference, both for the teeth inside (p<0.001) and outside the cleft area (p<0.001), with diffuse opacity (code 2) being more frequent in the incisal third and the combination of diffuse opacity and hypoplasia (code 7) in all the three thirds.

Statistically significant differences were also found between individuals born with bilateral and unilateral clefts (cleft lip only and cleft lip with cleft palate) for the grouped index of all incisors together, both for the teeth inside (p = 0.04) and outside the cleft area (p<0.001). The

**Table 3. Means and medians of the percentage of the surface area affected by the defect of the teeth inside and outside the cleft area, for all types of clefts analyzed.**

| | | Inside the cleft area | | Outside the cleft area | |
|---|---|---|---|---|---|
| Type of Cleft | N (individuals) | Mean | Median | Mean | Median |
| Unilateral cleft lip only | 19 | 43.1 | 33.6 | 22.2 | 14.3 |
| | | | | p<0.001 | |
| Bilateral cleft lip only* | 8 | 47.4 | 41.8 | - | - |
| Unilateral cleft lip with cleft palate | 130 | 44.2 | 39.8 | 30.1 | 23.9 |
| | | | | p<0.001 | |
| Bilateral cleft lip with cleft palate | 65 | 54.9 | 47.4 | 54.5 | 48.2 |
| | | | | p>0.05 | |
| Cleft palate only* | 11 | - | - | 33.6 | 25.5 |

Note:

*No cases with bilateral cleft lip only had teeth we considered to be outside the cleft area, since we only evaluated maxillary incisors. Similarly, no cleft palate only cases were affecting maxillary incisors.

difference is in the combinations of defects (codes 5, 6, and 7), which are more frequent in bilateral than in unilateral, showing that in bilateral cases there is a greater chance of finding more than one type of enamel defect on the buccal surface, both for teeth inside (p = 0.008016) and outside the cleft area (p = 0.000358), therefore exhibiting greater severity.

The frequency of individuals exhibiting teeth with developmental defects of enamel, concerning the cleft and non-cleft side, was statistically different (p<0.001). However, the averages of the percentage of the surface area affected by the defect for each tooth, inside and outside the cleft area, were similar (p = 0.18). Therefore, we cannot say that enamel defects on the cleft side are more extensive and severe than those on the non-cleft side. We observed that the frequency is, in fact, different (p<0.001; Odds Ratio = 7.15, 95% confidence interval: 4.674–10.942), in which a tooth outside the cleft area was 7.15 times more likely to be without defect in comparison to a tooth within the cleft area. However, when affected, the extent of the defects was similar, as shown in Table 4.

Regarding the type of cleft and the location of enamel defects on the buccal surface, statistically significant differences were found for the teeth outside the cleft area (p<0.001) and for

**Table 4. Frequency of individuals presenting teeth with and without the presence of developmental defects of enamel, inside and outside the cleft area.**

| | Frequency of teeth affected by individual (%Surface Affected Area) | | | | | |
|---|---|---|---|---|---|---|
| Location | Maxillary right lateral incisor (%SAD) | Maxillary right central incisor (%SAD) | Maxillary left central incisor (%SAD) | Maxillary left lateral incisor (%SAD) | Total | P-value |
| Inside the cleft area | 31 (53.57) | 71 (53.3) | 118 (46.7) | 49 (48.68) | 269 | **<0.001 (0.18)** |
| Outside the cleft area | 49 (25.67) | 77 (31.53) | 55 (43.71) | 18 (31.34) | 199 | |
| Total | 80 | 148 | 173 | 67 | 468 | |
| | Frequency of teeth not affected by individual | | | | | |
| Location | Maxillary right lateral incisor | Maxillary right central incisor | Maxillary left central incisor | Maxillary left lateral incisor | Total | |
| Inside the cleft area | 7 | 10 | 9 | 5 | 31 | 0.55 |
| Outside the cleft area | 50 | 56 | 30 | 28 | 164 | |
| Total | 57 | 66 | 39 | 33 | 195 | |

**Table 5. Frequency of teeth affected by the type of cleft concerning the location of the enamel defect on the buccal surface, inside and outside the cleft area (n = number of individuals evaluated).**

| Outside the cleft area | Type of Cleft | | | | | |
|---|---|---|---|---|---|---|
| Location of DDE | Bilateral cleft lip with cleft palate (n = 25) | Unilateral cleft lip with cleft palate (n = 83) | P-value* | Unilateral cleft lip only (n = 13) | Cleft palate only(n = 5) | Total |
| Cervical | 0 | 21 | 0.001 | 2 | 0 | 23 |
| Cervical/Incisal | 1 | 6 | 0.67 | 1 | 2 | 10 |
| Cervical/Middle | 2 | 17 | 0.24 | 1 | 1 | 21 |
| Cervical/Middle/Incisal | 19 | 27 | <0.001 | 1 | 2 | 49 |
| Incisal | 2 | 35 | 0.009 | 9 | 4 | 50 |
| Middle | 2 | 11 | 0.62 | 3 | 0 | 16 |
| Middle/Incisal | 8 | 14 | 0.04 | 7 | 1 | 30 |
| Total | 34 | 131 | <0.001 | 24 | 10 | 199 |
| Inside the cleft area | Type of Cleft | | | | | |
| Location of DDE | Bilateral cleft lip with cleft palate (n = 65) | Unilateral cleft lip with cleft palate (n = 130) | P-value | Unilateral cleft lip only (n = 19) | Bilateral cleft lip only (n = 8) | Total |
| Cervical | 8 | 8 | 0.71 | 1 | 1 | 18 |
| Cervical/Incisal | 5 | 4 | 1 | 1 | 2 | 12 |
| Cervical/Middle | 13 | 7 | 0.32 | 2 | 2 | 24 |
| Cervical/Middle/Incisal | 76 | 49 | 0.03 | 6 | 8 | 139 |
| Incisal | 8 | 21 | 0.001 | 4 | 1 | 34 |
| Middle | 5 | 7 | 0.36 | 1 | 1 | 14 |
| Middle/Incisal | 14 | 12 | 1 | 2 | 0 | 28 |
| Total | 129 | 108 | 0.05 | 17 | 15 | 269 |

those inside the cleft area (p = 0.022). Table 5 shows the frequency of affected teeth found in each third. It should be noted that the number of teeth and not individuals were used due to the small sample size since cleft lip with or without cleft palate and developmental defects of enamel are relatively rare conditions. Besides, teeth outside the cleft area are uncommon when considering bilateral clefts, in which generally the central and maxillary lateral incisors are adjacent to the cleft and, therefore, inside the cleft area, being less common the cases in which the cleft is located between the lateral incisor and canine.

In agreement with the results of Table 3, in Table 5 we can observe statistically significant differences (p<0.001) concerning the variable location of the enamel defect for the teeth outside the cleft area when comparing the bilateral and unilateral clefts. Individuals born with bilateral cleft lip were 7.85 times more likely to have more than one-third of the buccal surface affected (odds ratio = 7.85156, 95% confidence interval 2.619–23.542), and also presented, more frequently, enamel defects in all thirds (p<0.001), whereas in the unilateral individuals, the defects were restricted to only one third, being more frequent in the incisal (p = 0.009) and cervical (p = 0.001) thirds.

For the teeth inside the cleft area, the results found were corresponding, with unilateral individuals exhibiting a higher frequency of enamel defects in the incisal third (p = 0.001) while the bilateral individuals, in all thirds (p = 0.03), agreeing once again with the previous findings that, when affected, enamel defects are similarly found in the buccal surface, both in extension and in location, regardless of their position to the cleft.

**Table 6. Frequency of teeth affected by the type of cleft concerning the DDE code, outside the cleft area (n = number of individuals evaluated).**

| Outside the cleft area | Type of Cleft | | | | | | |
|---|---|---|---|---|---|---|---|
| DDE Code | Bilateral cleft lip with cleft palate[a] | Unilateral cleft lip with cleft palate[b] | Unilateral cleft lip only[c] | Cleft palate only[d] | P-value[*] (axb) | P-value[**] (axc) | Total |
| | (n = 25) | (n = 83) | (n = 13) | (n = 5) | | | |
| Code 1 | 0 | 10 | 4 | 0 | p = 0.02 | p = 0.003 | 14 |
| Code 2 | 6 | 55 | 12 | 4 | p = 0.008 | p = 0.008 | 77 |
| Code 3 | 11 | 28 | 3 | 1 | p = 0.17 | p = 0.08 | 43 |
| Code 4 | 2 | 8 | 2 | 0 | p = 1 | p = 0.71 | 12 |
| Code 5 | 4 | 7 | 1 | 0 | p = 0.18 | p = 0.30 | 12 |
| Code 6 | 0 | 1 | 0 | 0 | p = 0.49 | p = 1 | 1 |
| Code 7 | 7 | 20 | 2 | 4 | p = 0.45 | p = 0.20 | 33 |
| Code 8 | 4 | 2 | 0 | 0 | p = 0.004 | p = 0.03 | 6 |
| Total[***] | 34 | 131 | 24 | 9 | p = 0.04 | p = 0.26 | 198 |

[*]comparison between bilateral and unilateral cleft lip with cleft palate

[**]comparison between bilateral cleft lip with cleft palate and unilateral cleft lip only

[***]Teeth with Code 9 were not included for these comparisons. There were 251 teeth missing or not possible to score and 17 teeth with restorations.

When analyzing the type of cleft concerning the enamel defect, statistically significant differences were found for the teeth outside the cleft area (p = 0.031), in which a higher frequency of milder defects, such as isolated opacities (codes 1 and 2), was observed for the unilateral individuals. The presence of all combined defects (code 8) was observed for the individuals born with bilateral cleft lip, as shown in Table 6.

## MMP2 (rs9923304) association analysis

Individuals with genotypes analyzed (n = 124) were allocated to two groups. In group 1 (G1) are the individuals who presented all the teeth affected or at least two teeth with a percentage of the surface area affected by the defect greater than 20% (n = 67). The others were allocated to group 2 (G2) (n = 57), exhibiting one or no affected teeth or two with percentages below 20%. A higher frequency of the C allele was observed in G2 and a higher frequency of the less common T allele in G1 (p = 0.019843) (Table 7).

Raw measurements and genotypes for all participants are available as a supplemental file (S1 File).

## Discussion

Studies have attempted to define sub-phenotypes of oral clefts based on dental development [6, 9, 35]. Evidence suggests that individuals born with oral clefts have a higher frequency of DDE [7, 8, 10, 17, 19]. The reason for this higher prevalence rate of enamel defects in individuals born with CLP remains unclear. Several different etiological factors have been suggested as responsible, such as illness, trauma, nutrition, and metabolic conditions generally [19]. The

**Table 7. Frequency of C and T alleles found in individuals allocated to G1 and G2.**

| Alleles | G1 (n = 67) | G2 (n = 57) | P-value |
|---|---|---|---|
| C | 71 | 77 | p = 0.01 |
| T | 63 | 37 | |

present study supports the hypothesis that the most viable etiological factors for these defects may be the same as those for the cleft [8, 10, 36]. It is possible that these dental anomalies could represent an incomplete manifestation of the clefting process [19]. In a longitudinal study, Malanczuk et al. (1999) [10] observed that 19 patients had at least one permanent tooth retaining the same structural changes of the primary tooth in the dental enamel, therefore exhibiting corresponding pathological mechanisms both for disorders of the dental structure and for the development of the primary palate.

There are few reports on the diagnosis of DDE and the differential diagnosis is not simple because mistakes can be made in its assessment [15, 37]. Although direct clinical examination is a fast and cheap method, it has many disadvantages such as observer bias and effects of visual problems related to fatigue of the examiner [30]. Photography has been employed to assist the diagnosis of clinical examinations and increase the accuracy in detecting DDE, with most studies demonstrating high reliability [30, 31, 37–40]. The photographic method may facilitate blinded and repeated examinations [30, 37] and can be kept for future reassessment or application of different approaches or score systems [30, 31, 40]. However, technical sensitivity, inability to use touch and cost are some of its disadvantages of the use of photographs [30]. Some have suggested that the photographic method was much more sensitive than direct clinical examination in detecting DDE and was the best method for epidemiological studies [30, 37]. For these reasons, we analyzed intraoral photographs in this study, and developed a new method that facilitates the diagnosis of enamel defects and provides more information. To decrease the potential impact on variability, only one examiner obtained the measurements from the photographs. However, this approach has limited our ability to suggest the method can be reliably used by others.

Over the years, numerous score systems have been proposed for measuring enamel defects, causing further confusion in reporting results and making comparisons between these studies difficult [32]. The modified DDE index has been employed in several studies of enamel defects [8, 15, 18, 19, 37]. However, as the aim of this study was to understand not only the frequency of enamel defects found in individuals born with CLP but also to compare the severity and extent between the types of cleft and the affected and unaffected side, the new method was developed with the intention of providing these particular details.

The low correlation found in this study (r = 0.2748) concerning the percentage of the surface affected by the defect (proposed method) and the Modified DDE Index was expected. Some of the reasons [32] for the proposed modification of the used DDE Index [41] were to enable the assessment of the severity of the defects found and because the recording of data was time-consuming. However, although the variable is ordinal, it is not necessarily progressive, leading to confusion. Code 1, for example, represents demarcated opacity, while code 2, diffuse. Code 1 can affect the entire surface, whereas code 2 may be restricted to only one of the thirds of the tooth. Severity, in this case, is relative. Our proposed method, being a continuous variable, does not allow for this inconsistency, explaining and justifying the low correlation we found between the two analyses.

The present study found no significant differences in the prevalence of dental anomalies, similar to previous work [9, 42, 43], including enamel defects [8, 18], between male and female individuals born with CLP.

We analyzed the presence of enamel defects in the permanent maxillary incisors since studies have demonstrated that these teeth are the most commonly affected [8, 17, 19, 44]. We found a higher frequency of enamel defects on the cleft side, for both primary and permanent dentitions, compared with the noncleft side, in agreement with previously reported data [8, 19, 45, 46]. We believe that the frequency is, in fact, different, in which a tooth outside the cleft area was 7.15 times more likely to be healthy compared to one within. However,

when affected, the extent, severity, and location of the defects were similar. Our findings confirmed that there is a higher frequency of defects within the cleft area, though there are no previous studies that have assessed their extent compared to the non-cleft side. We managed to obtain these data through the proposed measurement method because when we use the codes of the modified DDE index, it tells only if there is an enamel defect and its type. Therefore, it was believed that teeth outside the cleft area were less affected in severity, when, in fact, they are only in frequency. A probable explanation is that although environmental factors may play some role, the genetic factors are decisive, since we observed the same severity of these defects both in teeth inside and outside the cleft area [18, 19]. However, the mechanism by which the cleft may interfere with tooth formation is not well understood [19, 36, 42, 45].

In this study, both for the teeth inside and outside the cleft area, we observed a pattern when comparing UCLP with BCLP individuals. The results found for the UCLP are in agreement with other studies, in which most defects affected less than one third of the crown, commonly the incisal third [17, 18]. Also, the occurrence of a single defect on the tooth surface was observed, both on the cleft and noncleft sides [17] with the highest frequencies for opacities [17, 18].

We observed that BCLP individuals are 7.85 times more likely to have more than one third of the tooth surface affected, showing, more often, combinations of enamel defects. In agreement with this result, Ruiz et al. (2013) found that bilateral cleft lip and palate had the highest prevalence of enamel defects (47.4%), followed by left unilateral cleft (33.8%) and right unilateral cleft (18.8%) [19]. Further, Wangsrimongkol et al. (2013) found that the most prevalent missing teeth were found in 70.7% of subjects in BCLP group [43]. Al Jamal et al. (2010) observed that BCLP subjects had significantly more microdontia, dilaceration and hypoplastic teeth than subjects with UCLP [42]. One hypothesis is that in these patients, treatments required are more extensive because of the severity of the malformation and this may extend the period during which the tooth is exposed to cariogenic factors [19]. However, Alam & Alfawzan (2020) [47] evaluated sella turcica bridging, an important landmark in the cranium on lateral cephalogram that helps to identify pathologies related to syndromes that affect the craniofacial region. Seven parameters related to Sella turcica morphology and skeletal malocclusion were analyzed and the authors found that BCLP individuals exhibit smaller values of all seven parameters as compared to all other CLP groups. This result extrapolates the hypotheses that correlate tooth formation with the clefting process, showing that it is necessary for studies that investigate the possible reasons for the higher prevalence and severity of anomalies, in general, in BCLP individuals.

The T allele of *MMP2* rs9923304 was associated with cases with more extensive enamel defects. MMP2 encodes an enzyme that degrades type IV collagen and is constitutively expressed by most connective tissue cells including endothelial cells, osteoblasts, fibroblasts, and myoblasts [20, 22]. Altered expression and activity levels of MMP2 is known to be associated with pathological states, especially those involving tissue remodeling [22].

Previous studies have demonstrated that genes involved in enamel development are associated with DDE [21, 48, 49]. Also, studies have shown the association of *MMP2* with formation and mineralization of dental tissues [23–25], and dental conditions such as periapical lesions [50], non-carious loss of extensive composite resin restorations [22], and talon cusp [51]. However, this was the first time this gene was specifically studied for DDE. In periodontal tissues, excessive fluoride intake increases MMP2 expression [52], but genetic variation of *MMP2* was not associated with dental fluorosis [27]. Therefore, genetic variation of MMP2 may be an additional factor leading to enamel defects that can be added to others such as previous orthodontic treatment and/or surgical treatment [53–55].

We implemented a cross-sectional cohort design, which may provide some advantages since the interval between the presence of oral clefts and its initial surgical repair is many years prior to maxillary incisors have fully erupted. However, some limitations to this type of design include nonignorable exiting, measurement error, and selection bias [56]. It is not possible to determine if the individuals included in the study had relatively less severe enamel defects, and individuals with more severe enamel defects were not included since they already underwent restorative treatment, which prevented us from determining the extent of the original surface defect. Although measurement error and selection bias are not exclusively likely in cross-sectional cohort designs, the time between oral clefts interventions start and when the maxillary incisors could be evaluated may have influenced the sample we finally had available for this study. An additional limitation was the limited number of observations for some subtypes of clefts, such as unilateral cleft lip (n = 19) or bilateral cleft lip (n = 8).

*MMP2* rs9923304 TT genotype could serve as the basis for a genomic approach to define risks for enamel defects in individuals born with CLP. Longitudinal studies in larger samples need to be conducted, as well as other genetic studies, for confirmation of the presented results. Our study has the limitation of not having evaluated *MMP2* expression during human enamel formation of the studied individuals.

In the near future, genetics might be used as a tool to screen children that will need biology-based preventive approaches. From the findings presented here, it was possible to observe that the frequency and severity of enamel defects is strongly related to the CLP phenotype. Therefore, the broadening of the cleft phenotype may allow for better gene-mapping efforts and provide more effective genetic counseling.

## Supporting information

**S1 File.**
(XLSX)

## Acknowledgments

Ionaria O. Assis, Bianca G.N. Cavalcante, and Mariana Bezamat helped with the *MMP2* genotyping.

## Author Contributions

**Conceptualization:** Adriana Modesto, Alexandre R. Vieira.

**Data curation:** Juliane R. Lavôr, Rosa Helena W. Lacerda.

**Formal analysis:** Juliane R. Lavôr.

**Funding acquisition:** Rosa Helena W. Lacerda, Adriana Modesto, Alexandre R. Vieira.

**Investigation:** Juliane R. Lavôr, Adriana Modesto, Alexandre R. Vieira.

**Methodology:** Adriana Modesto, Alexandre R. Vieira.

**Project administration:** Rosa Helena W. Lacerda, Alexandre R. Vieira.

**Resources:** Adriana Modesto, Alexandre R. Vieira.

**Supervision:** Alexandre R. Vieira.

**Validation:** Rosa Helena W. Lacerda.

**Writing – original draft:** Juliane R. Lavôr.

**Writing – review & editing:** Rosa Helena W. Lacerda, Adriana Modesto, Alexandre R. Vieira.

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
