## [Decision Letter · Decision Letter 0]

11 Nov 2020

PONE-D-20-32069

Maxillary Incisor Enamel Defects in Individuals Born with Cleft Lip/Palate

PLOS ONE

Dear Dr. Vieira,

Thank you for submitting your manuscript to PLOS ONE. After careful consideration, we feel that it has merit but does not fully meet PLOS ONE’s publication criteria as it currently stands. Therefore, we invite you to submit a revised version of the manuscript that addresses the points raised during the review process.

The expert reviewers raise many points that should addressed. The largest overall issue with the manuscript is interpretability currently as written. We encourage the authors to specifically focus on readability in a revision.

We look forward to receiving your revised manuscript.

Kind regards,

JJ Cray Jr., Ph.D.

Academic Editor

PLOS ONE

Journal Requirements:

Reviewers' comments:

Reviewer's Responses to Questions

**Comments to the Author**

1. Is the manuscript technically sound, and do the data support the conclusions?

Reviewer #1: Yes

Reviewer #2: Yes

2. Has the statistical analysis been performed appropriately and rigorously? 

Reviewer #1: Yes

Reviewer #2: I Don't Know

3. Have the authors made all data underlying the findings in their manuscript fully available?

Reviewer #1: Yes

Reviewer #2: Yes

4. Is the manuscript presented in an intelligible fashion and written in standard English?

Reviewer #1: No

Reviewer #2: Yes

5. Review Comments to the Author

Reviewer #1: Thank you for your submission on the important topic. In many areas in the paper there are sentences that are not clear, so some rewriting is required. There are two sets of Tables 1 and 2. I shall make comments and suggestions page by page

Abstract - this appears long, however i do not have a word count

second sentence line 5 - 'has', not 'have'

3rd last line page 1 - 'individuals with BCLP were 7.85..'

Introduction

3rd para

line 3 - There is, however, a gap

line 4 - the reason 'why' there is..

5th para

'enamel is softer and more porous'

The presence of a DDE only potentiates existing increased caries, it doesnt create caries risk i itself - this needs to be made clear

It would be preferential to use maxillary rather than upper

6th para 'It is' rather than 'Which are'

compare nor compared

7th para

line 6 - has not have

there is varied use of MMP2 in italics and not

8th para - the verb tenses vary - i prefer past

M&M

It is unclear how the sample size was determined - was this a convenience sample that was a time limited recruitment process?

para 1 - last word study not report

para 2

line 4 - surfaces not faces

stage 1

more details needed for ring flash used

What type of cheek retractors were used? were they sterilised between patients and how?

When was the quality of the image assessed to determine if it needed to be retaken? What metrics were used to asses quality?

stage 2

Image assessment - after the calibration exercise, were any images assessed by a second examiner to determine inter-examiner agreement and validate continuing correlation with the reference standard?

para 2

'of participants to any of the ..'

inability instead of impossibility

Table1 (in M&M) - this isnt the modified DDE index anymore as you have modified it. I suggest creating a name for your index

Stage 3

You mention 'total area' was measured - how was this achieved without some sort of measurement reference in the images? without a reference you cannot measure area, only proportion of area

para 2 - delete 'approximately 10% of the sample' - this is superfluous

Results

the first paragraph should be moved down one paragraph, it is out of order

Table 1 (results) and Table 2 (results) should be deleted and incorporated into the text (most of the information is there already)

CLP has already been defined

Table 3 would benefit by having statistical analysis results added

I suggest using the term 'teeth' rather than 'elements' throughout the text

in para starting 'As expected' - delete 'as expected' - this is results not discussion

In table 4 - no need for % with the numbers as the unit is %SAD

in para below Table 4 - fourth line - delete 'counting' and 'to find'. Also, maxillary rather than 'upper'

last sentence before Table 5 - 'between the lateral incisor and canine'

Tables 5 and 6 - no need for p= in the table as header is 'p value'

Para after table 5 - 'enamel defects in all thirds' (same for following paragraph)

Table 6 - where is code 9 data?

Discussion - first para - primary not deciduous

para 2 - there should be a better word than tactility

At this stage you should discuss that only 1 examiner/rater was used and the limitations this places on the validity of the results

Para 4 - the sentence starting 'however, the first can reach..' is unclear, especially the reference to 'one third'

para 6- delete 'studies have demonstrated' and primary teeth not deciduous

para 8 - sentence starting 'regarding studies on dental...' is unclear and needs rewriting

In the sentence starting one hypothesis - rehabilitation and risk is mentioned - these has no context at present

final sentence - add 'for' after necessary

there is some variability in the format of the references - e.g., some have months and issue numbers, some dont

Reviewer #2: - This manuscript describes the results of a cohort cross-sectional study performed on 233 individuals born with cleft lip and/or palate. The authors aimed to evaluate the frequency, location, severity, and extent of developmental enamel defects found in the maxillary incisors as well as understand their relationship with the cleft side. Furthermore, the authors addressed the hypothesis that developmental enamel defects can be influenced by variation in the MMP2 genes (rs9923304). A better understanding of the association between oral clefts and developmental enamel defects will inform dental practitioners in preventing, diagnosing, and treating dental complications that appear to be inevitable consequences of oral clefts.

- As a reviewer with expertise in the field of dental anomalies among individuals with oral clefts, I offer the following perspectives and suggestions that should be considered in the revision of this manuscript.

Introduction:

- Last paragraph: "Thus, the aim of this study was to investigate (the) characteristics of enamel defects and individuals born with oral clefts. " The word (the) is missing.

Methods:

- The study offers a clear description of the study objectives, outcomes, outcome assessment methods, and exclusion criteria. The study describes examiners' calibration and reports the Intraclass Correlation Coefficient.

- First paragraph: "233 individuals born with CLP were evaluated, with (a) mean age of 13.13 years (ranging from 6 to 35 years-old)." The word (a) is missing.

- Stage 1 – Intraoral photographs: "After (an) initial examination, the surfaces of the teeth were cleaned and dried and the appearance of the enamel was recorded using a digital camera (Canon EOS Rebel T5i, Ohta-ku, Tokyo, Japan), with standard lens (Canon EF 100 mm macro lens) and settings (ISO 6400, speed 1/125 and aperture F/25), always under the same flash (Macro Ring Flash Sigma) and natural lighting conditions. " The word (an) is missing.

- While the authors talk about syndromic/non-syndromic patterns of oral clefts in their introduction, they don't mention the syndromic status of their participants in the methods section.

- Furthermore, the authors need to shed the light on the history of surgical and orthodontic treatment among their participants as such treatment may have an effect on the development of enamel defects among individuals with oral clefts.

1. Carpentier, S, Ghijselings, E, Schoenaers, J, Carels, C, Verdonck, A. 2014. Enamel defects on the maxillary premolars in patients with cleft lip and/or palate: a retrospective case-control study. Eur Arch Paediatr Dent. 15(3):159–165.

2. Korolenkova, MV, Starikova, NV, Udalova, NV. 2019. The role of external aetiological factors in dental anomalies in non-syndromic cleft lip and palate patients. Eur Arch Paediatr Dent. 20(2):105–111.

3. Marzouk T, Alves IL, Wong CL, DeLucia L, McKinney CM, Pendleton C, Howe BJ, Marazita ML, Peter TK, Kopycka-Kedzierawski DT, Morrison CS, Malmstrom H, Wang H, Shope ET. Association between Dental Anomalies and Orofacial Clefts: A Meta-analysis. JDR Clin Trans Res. 2020 Oct 8:2380084420964795. DOI: 10.1177/2380084420964795. Epub ahead of print. PMID: 33030085.

- The examiner's calibration (Stage 2 – Determination of cleft and DDE phenotypes) is confusing to the reader. According to the study, the initial examination was performed by R.H.W.L and then the study mentions " To eliminate inter-examiner differences, intraoral photographs of all participants were examined by the same evaluator (J.R.L)". What do the authors mean by the word "same"? Who performed the primary evaluation?

- Stage 4 – DNA samples and genotyping: "A single nucleotide polymorphisms (SNP) in the MMP2 gene (rs9923304) was selected, considering disequilibrium linkage and gene structure." Remove the letter (s) from "A single nucleotide polymorphism(s)"

- Stage 5 – Statistical analysis: "In order to verify (the) normal distribution of the numerical variables, the Shapiro-Wilk test was applied, followed by (an) analysis of variance with the Student’s t-test and the Mann-Whitney test, in the cases of normal and non-normal distribution, respectively." The word (the) and (an) are missing.

- As many clinicians may not be familiar with the "cohort cross-sectional study design", it may be useful for the authors to report the limitations of such design. Refer to (Hudson JI, Pope HG Jr, Glynn RJ. The cross-sectional cohort study: an underutilized design. Epidemiology. 2005 May;16(3):355-9. DOI: 10.1097/01.ede.0000158224.50593.e3. PMID: 15824552)

- While the authors reported that the number of teeth and not individuals were used for outcome assessment due to the small sample size available. It should be highlighted that one of the limitations of the present study is the presence of a limited number of participants in the unilateral and bilateral cleft lip groups. (n=19 uCL; n=8 bCL)

- The authors should try to offer an explanation of why the means and medians of the percentage of the surface area affected by the defect of the elements inside the cleft palate area and outside the bilateral cleft lip area were zero.

Results:

- The results section is confusing to the reader who seeks more definitive conclusions.

Discussion

- The discussion section is well written.

- 8th paragraph: "Regarding studies on dental anomalies, Wangsrimongkol et al. (2013) found that the most prevalent missing teeth (was) found in 70.7% of subjects in BCLP group [43]." Replace (was) with (were).

- While the authors highlighted the significance of genetic factors as a possible cause for the development of enamel defects among individuals with oral clefts, it is important to point out that previous orthodontic and/or surgical treatment can also contribute.

1. Carpentier, S, Ghijselings, E, Schoenaers, J, Carels, C, Verdonck, A. 2014. Enamel defects on the maxillary premolars in patients with cleft lip and/or palate: a retrospective case-control study. Eur Arch Paediatr Dent. 15(3):159–165.

2. Korolenkova, MV, Starikova, NV, Udalova, NV. 2019. The role of external aetiological factors in dental anomalies in non-syndromic cleft lip and palate patients. Eur Arch Paediatr Dent. 20(2):105–111.

3. Marzouk T, Alves IL, Wong CL, DeLucia L, McKinney CM, Pendleton C, Howe BJ, Marazita ML, Peter TK, Kopycka-Kedzierawski DT, Morrison CS, Malmstrom H, Wang H, Shope ET. Association between Dental Anomalies and Orofacial Clefts: A Meta-analysis. JDR Clin Trans Res. 2020 Oct 8:2380084420964795. DOI: 10.1177/2380084420964795. Epub ahead of print. PMID: 33030085.

6. PLOS authors have the option to publish the peer review history of their article (what does this mean?). If published, this will include your full peer review and any attached files.

Reviewer #1: No

Reviewer #2: **Yes: **Tamer Marzouk

---

## [Author Response · Author response to Decision Letter 0]

11 Nov 2020

Point-by-point answers to the critique are below. All changes in the text are marked in yellow:

Reviewer #1: Thank you for your submission on the important topic. In many areas in the paper there are sentences that are not clear, so some rewriting is required. There are two sets of Tables 1 and 2. I shall make comments and suggestions page by page

Abstract - this appears long, however i do not have a word count

second sentence line 5 - 'has', not 'have'

RESPONSE: We made the grammatical correction asked.

3rd last line page 1 - 'individuals with BCLP were 7.85..'

RESPONSE: We made the requested correction.

Introduction

3rd para

line 3 - There is, however, a gap

line 4 - the reason 'why' there is..

5th para

'enamel is softer and more porous'

RESPONSE: We made these corrections as requested.

The presence of a DDE only potentiates existing increased caries, it doesnt create caries risk i itself - this needs to be made clear

RESPONSE: The sentence does not state DDE creates risk, but rather that its presence is a risk indicator. The sentence right before mentions “facilitating the development.” We are not making a claim of causal relationship.

It would be preferential to use maxillary rather than upper

RESPONSE: We made these changes as requested.

6th para 'It is' rather than 'Which are'

compare nor compared

RESPONSE: We changed to “it is” as requested. Since the studies are published, we left the statement that in them it was compared.

7th para

line 6 - has not have

there is varied use of MMP2 in italics and not

RESPONSE: We made the correction. MMP2 appears in italic fonts when it refers to the gene and non-italic when it refers to the protein.

8th para - the verb tenses vary - i prefer past

RESPONSE: We made the correction to the past tense as suggested.

M&M

It is unclear how the sample size was determined - was this a convenience sample that was a time limited recruitment process?

para 1 - last word study not report

RESPONSE: The sample as determined by using all the available material. We added a clarification as suggested. We also made the correction on the style requested.

para 2

line 4 - surfaces not faces

RESPONSE: We made the correction as requested.

stage 1

more details needed for ring flash used

What type of cheek retractors were used? were they sterilised between patients and how?

When was the quality of the image assessed to determine if it needed to be retaken? What metrics were used to asses quality?

RESPONSE: We added the details assuggested.

stage 2

Image assessment - after the calibration exercise, were any images assessed by a second examiner to determine inter-examiner agreement and validate continuing correlation with the reference standard?

RESPONSE: No.

para 2

'of participants to any of the ..'

inability instead of impossibility

RESPONSE: We made the corrections as requested.

Table1 (in M&M) - this isnt the modified DDE index anymore as you have modified it. I suggest creating a name for your index

RESPONSE: We incorporated the suggestion as stated.

Stage 3

You mention 'total area' was measured - how was this achieved without some sort of measurement reference in the images? without a reference you cannot measure area, only proportion of area

RESPONSE: That is what we meant. We clarified the text to address the point raised.

para 2 - delete 'approximately 10% of the sample' - this is superfluous

RESPONSE: It was deleted as requested.

Results

the first paragraph should be moved down one paragraph, it is out of order

RESPONSE: We made the suggested change.

Table 1 (results) and Table 2 (results) should be deleted and incorporated into the text (most of the information is there already)

RESPONSE: We deleted the tables as suggested.

CLP has already been defined

RESPONSE: We deleted the definition as suggested.

Table 3 would benefit by having statistical analysis results added

RESPONSE: We added the p-values as requested.

I suggest using the term 'teeth' rather than 'elements' throughout the text

RESPONSE: We made the requested change.

in para starting 'As expected' - delete 'as expected' - this is results not discussion

RESPONSE: It was deleted as suggested.

In table 4 - no need for % with the numbers as the unit is %SAD

RESPONSE: We made the correction as suggested.

in para below Table 4 - fourth line - delete 'counting' and 'to find'. Also, maxillary rather than 'upper'

last sentence before Table 5 - 'between the lateral incisor and canine'

RESPONSE: We made the suggested corrections.

Tables 5 and 6 - no need for p= in the table as header is 'p value'

RESPONSE: We made the corrections as suggested.

Para after table 5 - 'enamel defects in all thirds' (same for following paragraph)

RESPONSE: We made the suggested correction.

Table 6 - where is code 9 data?

RESPONSE: We added the missing information as requested.

Discussion - first para - primary not deciduous

RESPONSE: We made the correction as requested.

para 2 - there should be a better word than tactility

RESPONSE: We changed the word as requested.

At this stage you should discuss that only 1 examiner/rater was used and the limitations this places on the validity of the results

RESPONSE: We added the comment in the discussion as requested.

Para 4 - the sentence starting 'however, the first can reach..' is unclear, especially the reference to 'one third'

RESPONSE: We rewrote the sentence as suggested.

para 6- delete 'studies have demonstrated' and primary teeth not deciduous

RESPONSE: We made the suggested corrections.

para 8 - sentence starting 'regarding studies on dental...' is unclear and needs rewriting

In the sentence starting one hypothesis - rehabilitation and risk is mentioned - these has no context at present

final sentence - add 'for' after necessary

RESPONSE: We made the corrections as suggested.

there is some variability in the format of the references - e.g., some have months and issue numbers, some don’t

RESPONSE: We removed months as suggested.

Reviewer #2: - This manuscript describes the results of a cohort cross-sectional study performed on 233 individuals born with cleft lip and/or palate. The authors aimed to evaluate the frequency, location, severity, and extent of developmental enamel defects found in the maxillary incisors as well as understand their relationship with the cleft side. Furthermore, the authors addressed the hypothesis that developmental enamel defects can be influenced by variation in the MMP2 genes (rs9923304). A better understanding of the association between oral clefts and developmental enamel defects will inform dental practitioners in preventing, diagnosing, and treating dental complications that appear to be inevitable consequences of oral clefts.

- As a reviewer with expertise in the field of dental anomalies among individuals with oral clefts, I offer the following perspectives and suggestions that should be considered in the revision of this manuscript.

Introduction:

- Last paragraph: "Thus, the aim of this study was to investigate (the) characteristics of enamel defects and individuals born with oral clefts. " The word (the) is missing.

RESPONSE: We made the correction requested.

Methods:

- The study offers a clear description of the study objectives, outcomes, outcome assessment methods, and exclusion criteria. The study describes examiners' calibration and reports the Intraclass Correlation Coefficient.

- First paragraph: "233 individuals born with CLP were evaluated, with (a) mean age of 13.13 years (ranging from 6 to 35 years-old)." The word (a) is missing.

RESPONSE; We made the correction.

- Stage 1 – Intraoral photographs: "After (an) initial examination, the surfaces of the teeth were cleaned and dried and the appearance of the enamel was recorded using a digital camera (Canon EOS Rebel T5i, Ohta-ku, Tokyo, Japan), with standard lens (Canon EF 100 mm macro lens) and settings (ISO 6400, speed 1/125 and aperture F/25), always under the same flash (Macro Ring Flash Sigma) and natural lighting conditions. " The word (an) is missing.

RESPONSE: We made the correction requested.

- While the authors talk about syndromic/non-syndromic patterns of oral clefts in their introduction, they don't mention the syndromic status of their participants in the methods section.

RESPONSE: We added the information in the first paragraph of the methods section to address this comment.

- Furthermore, the authors need to shed the light on the history of surgical and orthodontic treatment among their participants as such treatment may have an effect on the development of enamel defects among individuals with oral clefts.

1. Carpentier, S, Ghijselings, E, Schoenaers, J, Carels, C, Verdonck, A. 2014. Enamel defects on the maxillary premolars in patients with cleft lip and/or palate: a retrospective case-control study. Eur Arch Paediatr Dent. 15(3):159–165.

2. Korolenkova, MV, Starikova, NV, Udalova, NV. 2019. The role of external aetiological factors in dental anomalies in non-syndromic cleft lip and palate patients. Eur Arch Paediatr Dent. 20(2):105–111.

3. Marzouk T, Alves IL, Wong CL, DeLucia L, McKinney CM, Pendleton C, Howe BJ, Marazita ML, Peter TK, Kopycka-Kedzierawski DT, Morrison CS, Malmstrom H, Wang H, Shope ET. Association between Dental Anomalies and Orofacial Clefts: A Meta-analysis. JDR Clin Trans Res. 2020 Oct 8:2380084420964795. DOI: 10.1177/2380084420964795. Epub ahead of print. PMID: 33030085.

RESPONSE: We added a comment regarding surgical and orthodontic treatment as suggested in the first paragraph. Photographic material is obtained prior to orthodontic treatment and many years after corrective cleft surgery for the lip. The references the reviewer listed also include changes in other teeth not studied by us.

- The examiner's calibration (Stage 2 – Determination of cleft and DDE phenotypes) is confusing to the reader. According to the study, the initial examination was performed by R.H.W.L and then the study mentions " To eliminate inter-examiner differences, intraoral photographs of all participants were examined by the same evaluator (J.R.L)". What do the authors mean by the word "same"? Who performed the primary evaluation?

RESPONSE: We edited the text to clarify this concern. R.H.W.L. did all photographic documentation and J.R.L. did all the measurements in the existing photographs.

- Stage 4 – DNA samples and genotyping: "A single nucleotide polymorphisms (SNP) in the MMP2 gene (rs9923304) was selected, considering disequilibrium linkage and gene structure." Remove the letter (s) from "A single nucleotide polymorphism(s)"

RESPONSE: We made the correction requested.

- Stage 5 – Statistical analysis: "In order to verify (the) normal distribution of the numerical variables, the Shapiro-Wilk test was applied, followed by (an) analysis of variance with the Student’s t-test and the Mann-Whitney test, in the cases of normal and non-normal distribution, respectively." The word (the) and (an) are missing.

RESPONSE: We made the corrections described.

- As many clinicians may not be familiar with the "cohort cross-sectional study design", it may be useful for the authors to report the limitations of such design. Refer to (Hudson JI, Pope HG Jr, Glynn RJ. The cross-sectional cohort study: an underutilized design. Epidemiology. 2005 May;16(3):355-9. DOI: 10.1097/01.ede.0000158224.50593.e3. PMID: 15824552)

RESPONSE: We added the limitations as suggested in the discussion section.

- While the authors reported that the number of teeth and not individuals were used for outcome assessment due to the small sample size available. It should be highlighted that one of the limitations of the present study is the presence of a limited number of participants in the unilateral and bilateral cleft lip groups. (n=19 uCL; n=8 bCL)

RESPONSE: We added this point as well in the discussion section.

- The authors should try to offer an explanation of why the means and medians of the percentage of the surface area affected by the defect of the elements inside the cleft palate area and outside the bilateral cleft lip area were zero.

RESPONSE: We added the requested explanation as a note to the table.

Results:

- The results section is confusing to the reader who seeks more definitive conclusions.

RESPONSE: The other reviewer suggested some edits that may help with this concern.

Discussion

- The discussion section is well written.

- 8th paragraph: "Regarding studies on dental anomalies, Wangsrimongkol et al. (2013) found that the most prevalent missing teeth (was) found in 70.7% of subjects in BCLP group [43]." Replace (was) with (were).

RESPONSE: We made the correction suggested.

- While the authors highlighted the significance of genetic factors as a possible cause for the development of enamel defects among individuals with oral clefts, it is important to point out that previous orthodontic and/or surgical treatment can also contribute.

1. Carpentier, S, Ghijselings, E, Schoenaers, J, Carels, C, Verdonck, A. 2014. Enamel defects on the maxillary premolars in patients with cleft lip and/or palate: a retrospective case-control study. Eur Arch Paediatr Dent. 15(3):159–165.

2. Korolenkova, MV, Starikova, NV, Udalova, NV. 2019. The role of external aetiological factors in dental anomalies in non-syndromic cleft lip and palate patients. Eur Arch Paediatr Dent. 20(2):105–111.

3. Marzouk T, Alves IL, Wong CL, DeLucia L, McKinney CM, Pendleton C, Howe BJ, Marazita ML, Peter TK, Kopycka-Kedzierawski DT, Morrison CS, Malmstrom H, Wang H, Shope ET. Association between Dental Anomalies and Orofacial Clefts: A Meta-analysis. JDR Clin Trans Res. 2020 Oct 8:2380084420964795. DOI: 10.1177/2380084420964795. Epub ahead of print. PMID: 33030085.

RESPONSE: We added a comment in the discussion and these references as suggested.

---

## [Decision Letter · Decision Letter 1]

9 Dec 2020

PONE-D-20-32069R1

Maxillary Incisor Enamel Defects in Individuals Born with Cleft Lip/Palate

PLOS ONE

Dear Dr. Vieira,

Thank you for submitting your manuscript to PLOS ONE. After careful consideration, we feel that it has merit but does not fully meet PLOS ONE’s publication criteria as it currently stands. Therefore, we invite you to submit a revised version of the manuscript that addresses the points raised during the review process.

Please consider some of the changes articulated by the second reviewer. This resubmission will be handled exclusively by the assigned AE.

We look forward to receiving your revised manuscript.

Kind regards,

JJ Cray Jr., Ph.D.

Academic Editor

PLOS ONE

Reviewers' comments:

Reviewer's Responses to Questions

**Comments to the Author**

1. If the authors have adequately addressed your comments raised in a previous round of review and you feel that this manuscript is now acceptable for publication, you may indicate that here to bypass the “Comments to the Author” section, enter your conflict of interest statement in the “Confidential to Editor” section, and submit your "Accept" recommendation.

Reviewer #1: All comments have been addressed

Reviewer #2: All comments have been addressed

2. Is the manuscript technically sound, and do the data support the conclusions?

Reviewer #1: Yes

Reviewer #2: Yes

3. Has the statistical analysis been performed appropriately and rigorously? 

Reviewer #1: Yes

Reviewer #2: I Don't Know

4. Have the authors made all data underlying the findings in their manuscript fully available?

Reviewer #1: (No Response)

Reviewer #2: Yes

5. Is the manuscript presented in an intelligible fashion and written in standard English?

Reviewer #1: Yes

Reviewer #2: Yes

6. Review Comments to the Author

Reviewer #1: Thank you for your responses, i believe you have addressed my comments adequately - you will probably find some minor amendments to the written language at final editing

Reviewer #2: Thank you to the authors for revising and resubmitting the manuscript on this important topic. There are a few sentences where rewriting could be beneficial.

Materials and Methods

The last sentence in the first paragraph "Photographic material was obtained prior orthodontic treatment and several years after surgical repair of the lip occurred."

- Please consider rephrasing to "Photographic material was obtained prior [to] orthodontic treatment and several years after [surgical lip repair]" for clarity.

The last sentence in the second paragraph "The exclusion criteria included labial surfaces of permanent central and lateral incisors not accessible for proper examination (presence of restorations, orthodontic appliances or crowns) or individuals with [bad] quality intraoral photographs."

- Please consider changing [bad] to [low]

Stage 1 – Intraoral photographs

"Cheeks and lips were retracted using [dental cheek lip retractor mouth opener T-shape], which was sterilized prior [to] each use."

-Please consider changing [dental cheek lip retractor mouth opener T-shape] to [T-Shape intraoral cheek lip retractor]

- Please consider adding [to]

"When not acceptable because it was not in focused according to the naked eye, the photograph was repeated."

- Please consider rephrasing to [When a photograph was not acceptable because of being out of focus, it was repeated]

Stage 2 – Determination of cleft and DDE phenotypes

Second to the last sentence in the second paragraph

"A new code (9) was added due to the inabibility of observing defects in some teeth, especially those adjacent to the cleft, where the tooth is often distalized, mesialized, ectopic, not erupted or absent"

- Please change [inabibility] to [inability]

"Table 1. Classification of defects that includes a modification of the modified DDE index, which [in] the addition of code 9."

- Please consider changing [in] to [includes]

"Table 3. Note: *No cases with bilateral cleft lip only had teeth we considered to be outside the cleft [área], since we only evaluated maxillary incisors. Similarly, no cleft palate only cases were affecting maxillary incisors."

- Please consider changing [área] to [area]

7. PLOS authors have the option to publish the peer review history of their article (what does this mean?). If published, this will include your full peer review and any attached files.

Reviewer #1: No

Reviewer #2: **Yes: **Tamer Marzouk

---

## [Author Response · Author response to Decision Letter 1]

9 Dec 2020

Point-by-point answers to the critique are below. All changes in the text are marked in yellow:

Reviewer #1: Thank you for your responses, i believe you have addressed my comments adequately - you will probably find some minor amendments to the written language at final editing

Reviewer #2: Thank you to the authors for revising and resubmitting the manuscript on this important topic. There are a few sentences where rewriting could be beneficial.

Materials and Methods

The last sentence in the first paragraph "Photographic material was obtained prior orthodontic treatment and several years after surgical repair of the lip occurred."

- Please consider rephrasing to "Photographic material was obtained prior [to] orthodontic treatment and several years after [surgical lip repair]" for clarity.

RESPONSE: We rewrote the text as suggested.

The last sentence in the second paragraph "The exclusion criteria included labial surfaces of permanent central and lateral incisors not accessible for proper examination (presence of restorations, orthodontic appliances or crowns) or individuals with [bad] quality intraoral photographs."

- Please consider changing [bad] to [low]

RESPONSE: We made the change as suggested.

Stage 1 – Intraoral photographs

"Cheeks and lips were retracted using [dental cheek lip retractor mouth opener T-shape], which was sterilized prior [to] each use."

-Please consider changing [dental cheek lip retractor mouth opener T-shape] to [T-Shape intraoral cheek lip retractor]

- Please consider adding [to]

RESPONSE: We made the suggested changes.

"When not acceptable because it was not in focused according to the naked eye, the photograph was repeated."

- Please consider rephrasing to [When a photograph was not acceptable because of being out of focus, it was repeated]

RESPONSE: We made the suggested change.

Stage 2 – Determination of cleft and DDE phenotypes

Second to the last sentence in the second paragraph

"A new code (9) was added due to the inabibility of observing defects in some teeth, especially those adjacent to the cleft, where the tooth is often distalized, mesialized, ectopic, not erupted or absent"

- Please change [inabibility] to [inability]

RESPONSE: We made the correction on the typo.

"Table 1. Classification of defects that includes a modification of the modified DDE index, which [in] the addition of code 9."

- Please consider changing [in] to [includes]

RESPONSE: We made the correction of the typo.

"Table 3. Note: *No cases with bilateral cleft lip only had teeth we considered to be outside the cleft [área], since we only evaluated maxillary incisors. Similarly, no cleft palate only cases were affecting maxillary incisors."

- Please consider changing [área] to [area]

RESPONSE: We made the correction of the typo.

---

## [Editor Report · Decision Letter 2]

11 Dec 2020

Maxillary Incisor Enamel Defects in Individuals Born with Cleft Lip/Palate

PONE-D-20-32069R2

Dear Dr. Vieira,

We’re pleased to inform you that your manuscript has been judged scientifically suitable for publication and will be formally accepted for publication once it meets all outstanding technical requirements.

Kind regards,

JJ Cray Jr., Ph.D.

Academic Editor

PLOS ONE
---

## [Editor Report · Acceptance letter]

15 Dec 2020

PONE-D-20-32069R2 

Maxillary Incisor Enamel Defects in Individuals Born with Cleft Lip/Palate 

Dear Dr. Vieira:

I'm pleased to inform you that your manuscript has been deemed suitable for publication in PLOS ONE. Congratulations! Your manuscript is now with our production department. 

Kind regards, 

on behalf of

Dr. JJ Cray Jr. 

Academic Editor

PLOS ONE